# Biomedical Named Entity Recognition via Dictionary-based Synonym Generalization

**Zihao Fu**♠   **Yixuan Su**♠,♡   **Zaiqiao Meng**♠◇   **Nigel Collier**♠

♠Language Technology Lab, University of Cambridge
♡ Cohere    ◇School of Computing Science, University of Glasgow
♠{zf268, nhc30}@cam.ac.uk
♡yixuan@cohere.com   ◇zaiqiao.meng@glasgow.ac.uk

## Abstract

Biomedical named entity recognition is one of the core tasks in biomedical natural language processing (BioNLP). To tackle this task, numerous supervised/distantly supervised approaches have been proposed. Despite their remarkable success, these approaches inescapably demand laborious human effort. To alleviate the need of human effort, dictionary-based approaches have been proposed to extract named entities simply based on a given dictionary. However, one downside of existing dictionary-based approaches is that they are challenged to identify concept synonyms that are not listed in the given dictionary, which we refer as the synonym generalization problem.

In this study, we propose a novel Synonym Generalization (SynGen) framework that recognizes the biomedical concepts contained in the input text using span-based predictions. In particular, SynGen introduces two regularization terms, namely, (1) a synonym distance regularizer; and (2) a noise perturbation regularizer, to minimize the synonym generalization error. To demonstrate the effectiveness of our approach, we provide a theoretical analysis of the bound of synonym generalization error. We extensively evaluate our approach on a wide range of benchmarks and the results verify that SynGen outperforms previous dictionary-based models by notable margins. Lastly, we provide a detailed analysis to further reveal the merits and inner-workings of our approach.[1]

## 1 Introduction

Biomedical Named Entity Recognition (BioNER) (Settles, 2004; Habibi et al., 2017; Song et al., 2021; Sun et al., 2021) is one of the core tasks in biomedical natural language processing (BioNLP). It aims to identify phrases that refer to biomedical entities, thereby serving as the fundamental component for numerous downstream BioNLP tasks (Leaman and Gonzalez, 2008; Kocaman and Talby, 2021).

Existing BioNER approaches can be generally classified into three categories, i.e. (1) supervised methods; (2) distantly supervised methods; and (3) dictionary-based methods. Supervised methods (Wang et al., 2019b; Lee et al., 2020; Weber et al., 2021) train the BioNER model based on large-scale human-annotated data. However, annotating large-scale BioNER data is expensive as it requires intensive domain-specific human labor. To alleviate this problem, distantly supervised methods (Fries et al., 2017; Zhang et al., 2021; Zhou et al., 2022) create a weakly annotated training data based on an in-domain training corpus. Nonetheless, the creation of the weakly annotated data still demands a significant amount of human effort (Fries et al., 2017; Wang et al., 2019a; Shang et al., 2020). For instance, the preparation of the in-domain training corpus could be challenging as the corpus is expected to contain the corresponding target entities. To this end, most existing methods (Wang et al., 2019a; Shang et al., 2020) simply use the original training set without the annotation as the in-domain corpus, which greatly limits their applicability to more general domains. In contrast to the supervised/distantly-supervised methods, dictionary-based methods are able to train the model without human-annotated data. Most of the existing dictionary-based frameworks (Aronson, 2001; Song et al., 2015; Soldaini and Goharian, 2016; Nayel et al., 2019; Basaldella et al., 2020) identify phrases by matching the spans of the given sentence with entities of a dictionary, thereby avoiding the need of extra human involvement or in-domain corpus. As human/expert involvement in the biomedical domain is usually much more expensive than in the general domain, in this paper, we focus our study on the dictionary-based method for the task of BioNER.

Although dictionary-based approaches do not re-

---

[1]The data and source code of this paper can be obtained from https://github.com/fuzihaofzh/BioNER-SynGen

quire human intervention or in-domain corpus, they suffer from the *synonym generalization problem*, i.e. the dictionary only contains a limited number of synonyms of the biomedical concepts that appear in the text. Therefore, if an entity synonym in the text is not explicitly mentioned by the dictionary, it cannot be recognized. This problem severely undermines the recall of dictionary-based methods as, potentially, a huge amount of synonyms are not contained in the dictionary.

To address the synonym generalization problem, we propose SynGen (**Syn**onym **Gen**eralization) — a novel framework that generalizes the synonyms contained in the given dictionary to a broader domain. Figure 1 presents an overview of our approach. (1) In the training stage, SynGen first samples the synonyms from a given dictionary as the positive samples. Meanwhile, the negative samples are obtained by sampling spans from a general biomedical corpus. Then, it fine-tunes a pre-trained model to classify the positive and negative samples. In particular, SynGen introduces two novel regularizers, namely a synonym distance regularizer which reduces the spatial distance; and a noise perturbation regularizer which reduces the gap of synonyms' predictions, to minimize the synonym generalization error. These regularizers make the dictionary concepts generalizable to the entire domain. (2) During the inference stage, the input text is split into several spans and sent into the fine-tuned model to predict which spans are biomedical named entities. To demonstrate the effectiveness of the proposed approach, we provide a theoretical analysis to show that both of our proposed regularizers lead to the reduction of the synonym generalization error.

We extensively test our approach on five well-established benchmarks and illustrate that SynGen brings notable performance improvements over previous dictionary-based models on most evaluation metrics. Our results highlight the benefit of both of our proposed regularization methods through detailed ablation studies. Furthermore, we validate the effectiveness of SynGen under the few-shot setup, notably, with about 20% of the data, it achieves performances that are comparable to the results obtained with a full dictionary.

In summary, our contributions are:

- We propose SynGen — a novel dictionary-based method to solve the BioNER task via synonym generalization.

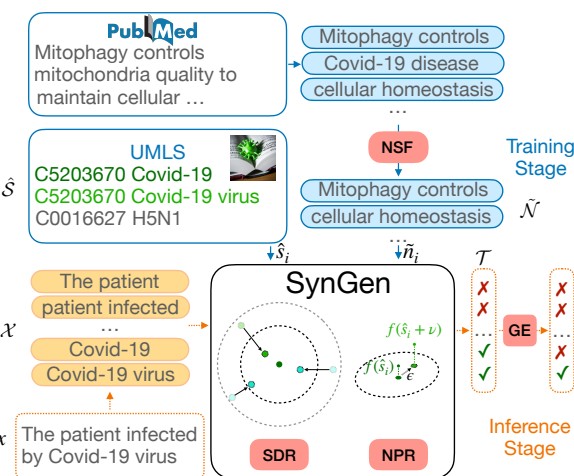

Figure 1: SynGen framework. $\rightarrow$ represents the training steps while $\dashrightarrow$ represents the inference steps.

- We provide a theoretical analysis showing that the optimization of SynGen is equivalent to minimizing the synonym generalization error.

- We conduct extensive experiments and analyses to demonstrate the effectiveness of our proposed approach.

## 2 Methodology

In this section, we first give the definition of the dictionary-based biomedical NER task. Then, we introduce our Synonym Generalization (i.e. SynGen) framework, following by the details of the synonym distance regularizer and the noise perturbation regularizer. Lastly, we provide a theoretical analysis on the problem of synonym generalization to show the effectiveness of our proposed approach.

### 2.1 Task Definition

Given a biomedical domain $\mathcal{D}$ (e.g. disease domain or chemical domain), we denote the set of all possible biomedical entities in $\mathcal{D}$ as $\mathcal{S} = \{\boldsymbol{s}_1, ..., \boldsymbol{s}_{|\mathcal{S}|}\}$, where $\boldsymbol{s}_i$ denotes the $i$-th entity and $|\mathcal{S}|$ denotes the size of $\mathcal{S}$. Then, given an input text $\boldsymbol{x}$, the task of biomedical NER is to identify subspans inside $\boldsymbol{x}$ that belong to $\mathcal{S}$, namely finding $\{\boldsymbol{x}_{[b_1:e_1]}, \cdots, \boldsymbol{x}_{[b_k:e_k]} | \forall i \in [1, k], \boldsymbol{x}_{[b_i:e_i]} \in \mathcal{S}\}$, where $k$ is the number of spans, and $b_i, e_i \in [1, |\boldsymbol{x}|]$ are the beginning and the ending token indices in $\boldsymbol{x}$ for the $i$-th span, respectively.

However, in real-life scenarios, it is impractical to enumerate all possible entities in $\mathcal{S}$. Normally, we only have access to a dictionary $\hat{\mathcal{S}} \subset \mathcal{S}$, $\hat{\mathcal{S}} = \{\hat{\boldsymbol{s}}_1, ..., \hat{\boldsymbol{s}}_{|\hat{\mathcal{S}}|}\}$ which contains a subset of

entities that belong to $\mathcal{S}$.[2] Thereby, the goal of dictionary-based biomedical NER is then to maximally recognize the biomedical entities from the input text conditioned on the available dictionary.

## 2.2 Synonym Generalization Framework

Figure 1 depicts an overview of the proposed Syn-Gen framework. (1) In the training stage (§2.2.1), it samples synonyms from a given dictionary (e.g. UMLS) as positive samples. Meanwhile, the negative samples are obtained by sampling spans from a general biomedical corpus (e.g. PubMed). Then, SynGen learns to classify positive and negative samples through the cross-entropy objective. Moreover, we propose two regularization methods (i.e. synonym distance regularization and noise perturbation regularization), which have been proved to be able to mitigate the synonym generalization error (detailed in §2.3). (2) In the inference stage (§2.2.1), SynGen splits the input text into different spans and scores them separately following a greedy extraction strategy.

### 2.2.1 Training

During training, we first sample a biomedical entity $\hat{s}_i$ (i.e. the positive sample) from the dictionary $\hat{\mathcal{S}}$ and encode its representation as $\hat{r}_i = E(\hat{s}_i)$, where $E(\cdot)$ is an encoder model such as BERT (Kenton and Toutanova, 2019). The probability of $\hat{s}_i$ being a biomedical entity is then modelled as

$$p(\hat{s}_i) = \sigma(\text{MLP}(\hat{r}_i)), \qquad (1)$$

where $\text{MLP}(\cdot)$ contains multiple linear layers and $\sigma(\cdot)$ is the sigmoid function. Meanwhile, we sample a negative text span $\tilde{n}_i$ from a general biomedical corpus, i.e. PubMed[3], and compute its representation as well as its probability as $\tilde{r}_i = E(\tilde{n}_i)$ and $p(\tilde{n}_i) = \sigma(\text{MLP}(\tilde{n}_i))$, respectively. Lastly, the training objective of our model is defined as

$$\mathcal{L}_c = -\frac{1}{2|\hat{\mathcal{S}}|} \sum_{i=0}^{|\hat{\mathcal{S}}|} \ln p(\hat{s}_i) + \ln[1 - p(\tilde{n}_i)]. \quad (2)$$

**Negative Sampling Filtering (NSF).** To obtain the negative sample $\tilde{n}_i$, we first sample spans with a random length from the PubMed corpus. Then, to avoid erroneously sampling the false negatives (i.e.

the spans that are biomedical entities), we encode all sampled spans into the embedding space and remove the samples that are close to the entities contained in the given dictionary. Specifically, we denote the set of sampled negative spans as $\tilde{\mathcal{N}}$. Then, $\forall \tilde{n}_i \in \tilde{\mathcal{N}}$, it satisfies

$$\min_{\forall \hat{s}_i \in \hat{\mathcal{S}}, \forall \tilde{n}_j \in \tilde{\mathcal{N}}} \|F(\hat{s}_i) - F(\tilde{n}_j)\| > t_d, \quad (3)$$

where $t_d$ is a hyper-parameter that specifies the threshold of minimal distance, and $F(\cdot)$ is an off-the-shelf encoder model. In our experiments, we use SAPBert (Liu et al., 2021a,b) as the model $F(\cdot)$ since it is originally designed to aggregate the synonyms of the same biomedical concept in the adjacent space.

**Synonym Distance Regularizer (SDR).** Intuitively, if the distinct synonyms of a single concept are concentrated in a small region, it is easier for the model to correctly identify them. Thereby, to equip our model with the ability to extract distinct synonyms of the same biomedical concept, we propose a novel regularization term—Synonym Distance Regularizer (SDR). During training, SDR first samples an anchor concept $\hat{s}_a$ and its associated concept $\hat{s}_p$ from the dictionary $\hat{\mathcal{S}}$.[4] Then, a random negative sample $\tilde{n}_n \in \tilde{\mathcal{N}}$ is sampled. Finally, SDR is computed by imposing a triplet margin loss (Chechik et al., 2010) between the encoded sampled synonyms and the sampled negative term as

$$\mathcal{R}_s = \max\{\|\hat{r}_a - \hat{r}_p\| - \|\hat{r}_a - \tilde{r}_n\| + \gamma_s, 0\}, \quad (4)$$

where $\gamma_s$ is a pre-defined margin, and $\hat{r}_a = E(\hat{s}_a)$, $\hat{r}_p = E(\hat{s}_p)$, and $\tilde{r}_n = E(\tilde{n}_n)$, respectively. In §2.3, we provide a rigorous theoretical analysis to show why reducing the distance between synonyms leads to minimizing the synonym generalization error.

**Noise Perturbation Regularizer (NPR).** Another way to mitigate the scoring gap between biomedical synonyms is to reduce the sharpness of the scoring function's landscape (Foret et al., 2020; Andriushchenko and Flammarion, 2022). This is because the synonyms of one biomedical entity are expected to distribute in a close-by region. Based on this motivation, we propose a new Noise Perturbation Regularizer (NPR) that is defined as

$$\mathcal{R}_n = \|p(\hat{s}_i + v) - p(\hat{s}_i)\|, \qquad (5)$$

---

[2]There are many biomedical dictionaries that are publicly available, such as UMLS (Bodenreider, 2004), Snomed CT (https://www.snomed.org), MeSH (https://meshb.nlm.nih.gov), etc.

[3]https://pubmed.ncbi.nlm.nih.gov/download/

[4]The $\hat{s}_a$ and $\hat{s}_p$ share the same concept ID.

where $\hat{s}_i$ is a biomedical entity sampled from the dictionary $\hat{S}$ and $v$ is a Gaussian noise vector. Intuitively, NPR tries to flatten the landscape of the loss function by minimizing the loss gap between vectors within a close-by region. More discussion of increasing the function flatness can be found in Foret et al. (2020); Bahri et al. (2022). In §2.3, we theoretically show why NPR also leads to the reduction of synonym generalization error.

**Overall Loss Function.** The overall learning objective of our SynGen is defined as

$$\mathcal{L} = \mathcal{L}_c + \alpha \mathcal{R}_s + \beta \mathcal{R}_n, \tag{6}$$

where $\alpha$ and $\beta$ are tunable hyperparameters that regulate the importance of the two regularizers.

### 2.2.2 Inference

During inference, given the input text $x$, SynGen first splits $x$ into spans with different lengths, namely, $\mathcal{X} = \{x_{[i:j]} | 0 \le i \le j \le |x|, j - i \le m_s\}$, where $m_s$ is the maximum length of the span. Then, we compute the score of every span $x_{[i:j]} \in \mathcal{X}$ as $p(x_{[i:j]})$ with Equation (1). We select the spans whose score is higher than a pre-defined threshold $t_p$ as candidates, which are then further filtered by the greedy extraction strategy as introduced below.

**Greedy Extraction (GE).** It has been observed that a lot of biomedical terms are nested (Finkel and Manning, 2009; Marinho et al., 2019). For example, the entity *T-cell prolymphocytic leukemia* contains sub-entities *T-cell* and *leukemia*. However, in most of the existing BioNER approaches, if one sentence $x$ contains the entity *T-cell prolymphocytic leukemia*, these approaches usually only identify it as a single biomedical entity and ignore its sub-entities *T-cell* or *leukemia*. To address this issue, our SynGen applies a greedy extraction (GE) strategy to post-process the extracted biomedical terms. In particular, our GE strategy first ranks the recognized terms by their length in descending orders as $\mathcal{T} = \{t_1, t_2, \cdots, t_n | \forall i < j, |t_i| > |t_j|\}$ and set the initial validation sequence $x^{(1)} = x$. Then, it checks the ranked terms from $t_1$ to $t_n$. If the term $t_i$ is a sub-sequence of the validation sequence $x^{(i)}$ (i.e. $\exists p, q < |x^{(i)}|$, such that $t_i = x_{[p:q]}^{(i)}$), it will recognize the term $t_i$ as a biomedical entity and set a new validation sequence $x^{(i+1)}$ by removing all $t_i$'s occurrence in the sequence $x^{(i)}$. As a result, the sub-entities inside a recognized entity will never be recognized again because they will not be contained in the corresponding validation sequence.

### 2.3 Theoretical Analysis

Most of the existing dictionary-based frameworks suffer from a common problem that terms outside of the given dictionary cannot be easily recognized, which we refer to as the *synonyms generalization problem*. To understand why the SynGen framework can resolve this problem, we give a theoretical analysis focusing on the correctness of entities in $\mathcal{S}$. Specifically, given a bounded negative log-likelihood loss function $f(r) = -\ln \sigma(\text{MLP}(r)) \in [0, b], r = E(s)$, it tends to 0 if an entity $s \in \mathcal{S}$ is correctly classified as positive. Otherwise, $f(r)$ tends to $b$ if the entity $s$ is wrongly recognized as a non-biomedical phrase. Following the traditional generalization error (Shalev-Shwartz and Ben-David, 2014), we further define the average empirical error for entities in the dictionary $\hat{S}$ as $\hat{R} = \frac{1}{|\hat{S}|} \sum_{i=1}^{|\hat{S}|} f(\hat{r}_i)$. To better analyze the generalization error for all synonyms, we consider the most pessimistic error gap between $\hat{R}$ and the error of arbitrary $s \in \mathcal{S}$, namely $f(E(s))$. Then, the synonym generalization error can be defined as follows:

**Definition 1** (synonym generalization error). *Given a loss function $f(r) \in [0, b]$, the synonym generalization error is defined as:*

$$E_s = \sup_{s \in \mathcal{S}} (f(E(s)) - \hat{R}).$$

It can be observed from Definition 1 that small $E_s$ implies error $f(E(s))$ for arbitrary $s$ will no deviate too far from $\hat{R}$. Therefore, training $f$ with the dictionary terms $\hat{S}$ is generalizable to the entities in the whole domain $\mathcal{S}$. To give a theoretical analysis of $E_s$, we further assume that $\hat{S}$ is an $\epsilon-$net of $\mathcal{S}$, namely, $\forall s \in \mathcal{S}, \exists \hat{s} \in \hat{S}$, such that $\|\hat{s} - s\| \le \epsilon$. Intuitively, given an entity $s \in \mathcal{S}$, we can always find an entity $\hat{s}$ in the dictionary $\hat{S}$ within distance $\epsilon$. We further assume that $f$ is $\kappa$-Lipschitz, namely, $\|f(x) - f(y)\| \le \kappa \|x - y\|$. Then, we have the following bound hold.

**Theorem 1** (Synonym Generalization Error Bound). *If $\hat{S}$ is an $\epsilon-$net of $\mathcal{S}$. The loss function $f \in [0, b]$ is $\kappa$-Lipschitz continuous. We have with probability at least $1 - \delta$,*

$$E_s < (\kappa\epsilon + b)\sqrt{\frac{\ln |\mathcal{S}| + \ln \frac{2}{\delta}}{2}} + b\sqrt{\frac{\ln \frac{2}{\delta}}{2|\hat{S}|}}. \tag{7}$$

The proof can be found in Appendix A. It can be observed from Theorem 1 that reducing the synonym distance upper bound $\epsilon$ or function $f$'s Lipschitz constant $\kappa$ can help reduce the generalization error gap $E_s$.

Theorem 1 explains why both SDR and NPR are able to help improve the NER performance. (1) For SDR, it allows the model to learn to reduce the distance between synonyms, which is equivalent to reducing $\epsilon$ of Equation (7). Therefore, it helps to reduce the synonym generalization error upper bound. (2) For NPR, it helps reducing the Lipschitz constant $\kappa$ because given a vector $v$, minimizing $\mathcal{R}_n$ is equivalent to minimizing

$$\frac{\|f(\hat{x}_i + v) - f(\hat{x}_i)\|}{\|v\|} = \frac{\|f(\hat{x}_i + v) - f(\hat{x}_i)\|}{\|(\hat{x}_i + v) - \hat{x}_i\|},$$

as $v$ is fixed during the parameter optimization procedure. Therefore, optimizing $\mathcal{R}_n$ is a necessary condition for reducing $f$'s Lipschitz constant $\kappa$.

## 3 Experiments

### 3.1 Experimental Setup

We evaluate all the models on 6 popular BioNER datasets, including two in the disease domain (i.e. NCBI disease (Doğan et al., 2014) and BC5CDR-D (Li et al., 2016)), two in the chemical domain (i.e. BC4CHEMD (Krallinger et al., 2015) and BC5CDR-C (Li et al., 2016)) and two in the species domain (i.e. Species-800 (Pafilis et al., 2013) and LINNAEUS (Gerner et al., 2010)). Note that BC5CDR-D and BC5CDR-C are splits of the BC5CDR dataset (Li et al., 2016) for evaluating the capability of recognizing entities in the disease and chemical domains respectively, following Lee et al. (2020). We evaluate the performance by reporting the Precision (P), Recall (R), and $F_1$ scores. The entity name dictionary used in our model is extracted by the concepts' synonyms in diseases, chemicals, and species partition from UMLS (Bodenreider, 2004). The negative spans are randomly sampled from the PubMed corpus [3].

We tune the hyper-parameters of SDR (i.e. $\alpha$), NPR (i.e. $\beta$), threshold constant $t_d$, $t_p$ and maximal span length $m_s$ by using grid search on the development set and report the results on the test set. The hyper-parameter search space as well as the best corresponding setting can be found in Appendix D. We use PubMedBert (Gu et al., 2021) as the backbone model by comparing the development set results from PubMedBert, SAPBert (Liu et al.,

2021a,b), BioBert (Lee et al., 2020), SciBert (Beltagy et al., 2019), and Bert (Kenton and Toutanova, 2019) (Detailed comparison can be found in Appendix C). Our experiments are conducted on a server with NVIDIA GeForce RTX 3090 GPUs. For all the experiments, we report the average performance of our used three metrics over 10 runs.

### 3.2 Comparison Models

We compare our model with baseline models depending on different kinds of annotations or extra efforts. The supervised models (BioBert and SBM) require golden annotations. The distantly supervised models (SBMCross, SWELLSHARK and AutoNER) depend on several different kinds of annotation or extra efforts which will be discussed in the corresponding models. The dictionary-based models mainly use the UMLS dictionary.

**BioBert** (Lee et al., 2020) first pre-trains an encoder model with biomedical corpus and then fine-tunes the model on annotated NER datasets.

**SBM** is a standard Span-Based Model (Lee et al., 2017; Luan et al., 2018, 2019; Wadden et al., 2019; Zhong and Chen, 2021) for NER task. We use the implementation by Zhong and Chen (2021).

**SBMCross** utilizes the same model as SBM. We follow the setting of Langnickel and Fluck (2021) to train the model on one dataset and test it on the other dataset in the same domain which is referred as the in-domain annotation. For example, in the NCBI task, we train the model on the BC5CDR-D dataset with SBM and report the results on the NCBI test set.

**SWELLSHARK** (Fries et al., 2017) proposes to first annotate a corpus with weak supervision. Then it uses the weakly annotated dataset to train an NER model. It requires extra expert effort for designing effective regular expressions as well as hand-tuning for some particular special cases.

**AutoNER** (Wang et al., 2019a; Shang et al., 2020) propose to first train an AutoPhase (Shang et al., 2018) tool and then tailor a domain-specific dictionary based on the given in-domain corpus. The corpus is the original training set corpus without human annotation. Afterwards, it trains the NER model based on the distantly annotated data. We also report the ablation experiments by removing the dictionary tailor or replacing the in-domain corpus with evenly sampled PubMed corpus.

**EmbSim** first uses a pre-trained model to encode the input spans and the entities in the UMLS dictio-

| Model | NCBI | | | BC5CDR-D | | | BC5CDR-C | | | BC4CHEMD | | | Species-800 | | | LINNAEUS | | | AVG | | |
|---|---|---|---|---|---|---|---|---|---|---|---|---|---|---|---|---|---|---|---|---|---|
| | P | R | $F_1$ | P | R | $F_1$ | P | R | $F_1$ | P | R | $F_1$ | P | R | $F_1$ | P | R | $F_1$ | P | R | $F_1$ |
| BioBert[♮] | 88.2 | 91.2 | 89.7 | 86.5 | 87.8 | 87.2 | 93.7 | 93.3 | 93.5 | 92.8 | 91.9 | 92.4 | 72.8 | 75.4 | 74.1 | 90.8 | 85.8 | 88.2 | 87.5 | 87.6 | 87.5 |
| SBM[♮] | 88.4 | 88.9 | 88.6 | 83.4 | 86.4 | 84.9 | 93.2 | 93.6 | 93.4 | 92.0 | 86.6 | 89.2 | 99.5 | 91.6 | 95.4 | 99.5 | 80.1 | 88.9 | 92.8 | 87.9 | 90.1 |
| SBMCross[◇] | 75.9 | 58.3 | 66.0 | 70.1 | 61.3 | 65.4 | 94.1 | 86.4 | 90.1 | 72.2 | 63.2 | 67.4 | 64.2 | 64.5 | 64.3 | 78.8 | 45.8 | 57.9 | 75.9 | 63.2 | 68.5 |
| SWELLSHARK[♭△] | 64.7 | 69.7 | 67.1 | 80.7 | 77.6 | 79.1 | 88.3 | 88.3 | 88.3 | - | - | - | - | - | - | - | - | - | 77.9 | 78.5 | 78.2 |
| AutoNER[♡♯] | 79.4 | 72.0 | 75.5 | 86.2 | 67.9 | 76.0 | 85.2 | 84.2 | 84.7 | 91.1 | 18.9 | 31.3 | 86.6 | 90.9 | 88.7 | 92.1 | 95.6 | 93.8 | 86.8 | 71.6 | 75.0 |
| AutoNER w/o DT[♡] | 66.8 | 32.4 | 43.6 | 72.0 | 17.3 | 27.9 | 89.7 | 67.3 | 76.9 | 90.7 | 19.7 | 32.4 | 57.6 | 50.7 | 53.9 | 88.4 | 39.0 | 54.1 | 77.5 | 37.7 | 48.1 |
| AutoNER w/o IDC[♯] | 85.1 | 19.1 | 31.2 | 87.1 | 40.4 | 55.2 | 94.2 | 37.3 | 53.4 | 91.2 | 18.8 | 31.2 | 83.6 | 18.5 | 30.3 | 90.4 | 62.8 | 74.1 | 88.6 | 32.8 | 45.9 |
| AutoNER w/o DT+IDC | 57.9 | 9.7 | 16.6 | 63.0 | 13.9 | 22.8 | 92.8 | 39.3 | 55.2 | 60.9 | 24.6 | 35.1 | 59.8 | 25.0 | 35.3 | 80.1 | 33.0 | 46.8 | 69.1 | 24.2 | 35.3 |
| EmbSim | 56.7 | 24.9 | 34.6 | 61.8 | 14.3 | 23.2 | 71.7 | 61.2 | 66.0 | 47.4 | 24.7 | 32.4 | 49.0 | 34.2 | 40.3 | 80.4 | 42.9 | 55.9 | 61.2 | 33.7 | 42.1 |
| MetaMap | 61.8 | 27.8 | 38.4 | 69.3 | 13.3 | 22.3 | 65.9 | 63.5 | 64.7 | 33.1 | 25.2 | 28.6 | 56.9 | 48.7 | 52.5 | 85.5 | 44.3 | 58.3 | 62.1 | 37.1 | 44.1 |
| MetaMap (Uncased) | 58.4 | 27.5 | 37.4 | 63.5 | 18.4 | 28.6 | **94.8** | 64.1 | 76.5 | **86.2** | 24.0 | 37.5 | 49.1 | 52.3 | 50.6 | 79.1 | 49.6 | 61.0 | 71.9 | 39.3 | 48.6 |
| SPED | 59.3 | 30.1 | 39.9 | 68.2 | 14.3 | 23.7 | 65.6 | 63.9 | 64.8 | 33.0 | 25.4 | 28.7 | 56.0 | 49.4 | 52.5 | 85.3 | 44.7 | 58.7 | 61.2 | 38.0 | 44.7 |
| TF-IDF | 26.1 | 29.7 | 27.7 | 32.0 | 22.6 | 26.4 | 74.1 | 65.4 | 69.5 | 19.1 | 39.3 | 25.7 | 42.5 | 21.4 | 28.4 | 77.3 | 40.5 | 53.1 | 45.2 | 36.5 | 38.5 |
| QuickUMLS | **80.4** | 17.2 | 28.4 | **93.5** | 14.5 | 25.1 | 93.2 | 56.9 | 70.7 | 82.7 | 16.9 | 28.1 | **61.7** | 46.7 | 53.2 | **88.2** | 44.7 | 59.3 | **83.3** | 32.8 | 44.1 |
| SynGen | 68.8 | **64.1** | 66.2 | 63.8 | **63.4** | 63.5 | 85.0 | **83.9** | 84.4 | 56.4 | **51.1** | 53.6 | 58.8 | **65.7** | 62.0 | 84.9 | **66.2** | 74.4 | 69.6 | **65.7** | 67.4 |

Table 1: Main results. We repeat each experiment for 10 runs and report the averaged scores. For BioBert and SWELLSHARK, we report the score from the original paper. We mark the extra effort involved with superscripts, where ♮ is gold annotations; ◇ is in-domain annotations; ♭ is regex design; △ is special case tuning; ♡ is in-domain corpus; ♯ is dictionary tailor. The bold values indicate the best performance among the dictionary-based models. The standard deviation analysis is in Figure 7.

nary into the corresponding vector representation. Then, it calculates the minimal distance between a given span and an arbitrary entity in the dictionary. It recognizes spans with minimal distances smaller than a threshold as named entities. We report the performance based on BioBert. For the results using other backbone models, please refer to Appendix C.

**MetaMap** (Aronson, 2001; Divita et al., 2014; Soldaini and Goharian, 2016) proposes to perform exact concept mapping of spans with the entities in the UMLS dictionary. We report the results for both cased and uncased matching.

**SPED** (Rudniy et al., 2012; Song et al., 2015) calculates the Shortest Path Edit Distances between the query span and each entity in the dictionary. Then it recognizes a span as a biomedical entity if the minimal distance ratio is smaller than a specific threshold.

**TF-IDF** follows the model of Ujiie et al. (2021). We remove the component utilizing the annotated data and only keep the tf-idf features to make it comparable to other models without extra effort.

**QuickUMLS** (Soldaini and Goharian, 2016) is a fast approximate UMLS dictionary matching system for medical entity extraction. It utilizes Simstring (Okazaki and Tsujii, 2010) as the matching model. We report the scores based on Jaccard distance. For the performance using other distances, please refer to Appendix C.

### 3.3 Experimental Results

**Main Results.** We first compare the overall performance of our proposed SynGen framework with the baseline models, and the results are shown in Table 1. It can be observed that: (1) Our proposed SynGen model outperforms all other dictionary-based models in terms of $F_1$ score. This is because it alleviates the synonym generalization problem and can extract entities outside of the dictionary. As a result, the recall scores are significantly improved. (2) By comparing SBM and SBMCross, we can find that the performance is very sensitive to the selection of the in-domain corpus. Even using the golden annotation of another training set in the same domain can lead to a sharp performance decrease. Therefore, preparing a good in-domain corpus is quite challenging. (3) SynGen is already comparable to the performance of SBMCross with average $F_1$ scores of 67.4 and 68.5 respectively. It shows that our dictionary-based model is comparable to a supervised model without in-domain data. (4) The precision score (P) of QuickUMLS is very high. This is because it is mainly based on matching exact terms in the dictionary and may not easily make mistakes. However, it cannot handle the synonyms out of the UMLS dictionary. As a result, it fails to retrieve adequate synonyms, leading to low recall scores. (5) By comparing the ablation experiments for AutoNER, we can conclude that the human labor-intensive dictionary tailor and the in-domain corpus selection are very important for distantly supervised methods. Without the dictionary tailor or the in-domain corpus,

| | NCBI | BC5C DR-D | BC5C DR-C | BC4C HEMD | Speci es-800 | LINN AEUS | AVG |
|---|---|---|---|---|---|---|---|
| SynGen | **66.2** | **63.5** | **84.4** | **53.6** | **62.0** | **74.4** | **67.4** |
| w/o SDR | 66.2 | 63.1 | 80.8 | 51.1 | 60.6 | 73.0 | 65.8 |
| w/o NPR | 66.2 | 62.8 | 78.0 | 49.7 | 60.3 | 73.1 | 65.0 |
| w/o NPR+SDR | 64.7 | 58.7 | 76.9 | 49.0 | 59.7 | 72.0 | 63.5 |
| w/o NSF | 60.3 | 49.5 | 76.8 | 49.0 | 54.3 | 54.7 | 57.4 |
| w/o GE | 49.8 | 54.4 | 69.4 | 33.4 | 52.1 | 71.7 | 55.1 |

Table 2: Ablation study[5].

| Backbone of SynGen | NCBI | BC5C DR-D | BC5C DR-C | BC4C HEMD | Speci es-800 | LINN AEUS | AVG |
|---|---|---|---|---|---|---|---|
| PubMedBert | 66.2 | **63.5** | **84.4** | **53.6** | **62.0** | **74.4** | **67.4** |
| SAPBert | **66.3** | 62.1 | 83.8 | 42.4 | 58.1 | 71.6 | 64.0 |
| BioBert | 62.4 | 57.4 | 81.0 | 31.7 | 52.4 | 59.3 | 57.4 |
| Bert | 62.0 | 56.1 | 79.6 | 47.1 | 55.2 | 62.1 | 60.4 |
| SciBert | 61.6 | 55.7 | 81.0 | 49.3 | 53.1 | 63.7 | 60.7 |

Table 3: Comparison of different backbone models[5].

the performance drops sharply. It should also be noted that our model outperforms the AutoNER model without in-domain corpus or dictionary tailoring significantly. It shows the effectiveness of our model without using extra effort (e.g. tailored dictionary).

**Ablation Study.** In order to show the effectiveness of our model's each component, we conduct several ablation experiments over different variants of our model, and the results are shown in Table 2 and Table 3. We have the following observations from the two tables. (1) By comparing the vanilla SynGen model with its variants, such as SynGen w/o NPR, SynGen w/o SDR, and SynGen w/o NPR + SDR, we can observe that removing any component results in performance drop, which means that our proposed NPR and SDR component can effectively improve the performance. This observation further verifies the correctness of our theory in Theorem 1. (2) By comparing SynGen with SynGen w/o NSF, we can find that the negative sample filtering component is also quite important in finding proper negative samples. (3) By comparing SynGen with SynGen w/o GE model, we also observe a significant performance drop, which concludes that the greedy extraction strategy does help improve the overall performance. Specifically, it helps improve the precision as it will not extract sub-entities inside other entities and thus avoid false positive extractions. (4) We also try to use different backbone models including PubMedBert, SAPBert, BioBert, Bert, and SciBert and the results are shown in Table 3. Our experiments show that PubMedBert performs the best. These results may be caused by the different training corpus used by each pre-trained model.

**Influence of Synonym Distance Regularizer.** To show that regularizing the synonym distance

by the SDR component does help improve the overall performance, we plot how the synonym distance changes as the SDR weight increases in Figure 6. Specifically, we train the model with different hyper-parameter $\alpha$ and measure the synonym distance by sampling 10,000 synonym pairs from UMLS. Then, we calculate the average distance between synonym name pairs for different $\alpha$. As suggested in Figure 6, as $\alpha$ increases, the synonym distance decreases which shows the effectiveness of the SDR component in controlling the synonym distance. On the other hand, we also plot how the evaluation scores change as the synonym distance increases in Figure 2. For the previous results, we further split the distance intervals into 8 segments and get the average overall performance in each interval. The results indicate that as the synonym distance is regularized (decreases), the overall performance increases. This observation shows the effectiveness of our proposed SDR component and justifies the analysis in Theorem 1.

**Influence of Noise Perturbation.** To show the usefulness of our proposed NPR component, we plot how the scores change as the NPR weight (i.e. $\beta$) changes. The results are shown in Figure 3. It indicates that as $\beta$ increases, all metrics including precision, recall, and $F_1$ scores increase. This observation shows that the NPR component does help improve the performance which also justifies our theoretical analysis.

**Few-Shot Analysis.** To further show that our SynGen framework can be applied in the few-shot scenarios, we run the model with part of the original dictionary entries with the dictionary size ratio ranging in $\{0.1\%, 0.2\%, 0.5\%, 1\%, 2\%, 5\%, 10\%, 20\%, 30\%, 40\%, 50\%, 60\%, 70\%, 80\%, 90\%, 100\%\}$. The results are shown in Figure 4. To better show the capability of few-shot learning, we also draw the same figure for the MetaMap model as shown in Figure 5. It can be concluded from the results that in SynGen, when the dictionary size is very

---

[5]The full results including precision and recall results can be found in Appendix C

[5]The full results including precision and recall can be found in Appendix C

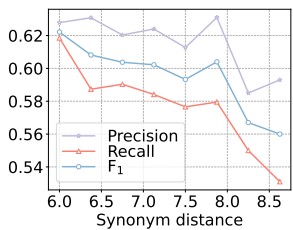

Figure 2: Influence of synonym distance.

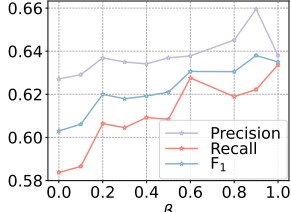

Figure 3: Influence of NPR.

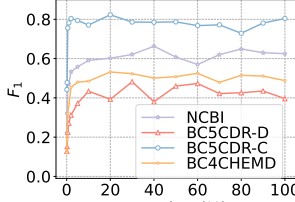

Figure 4: Few-shot analysis.

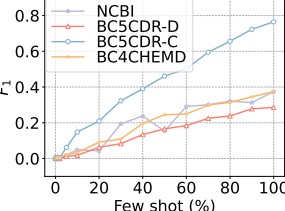

Figure 5: Few-shot analysis for MetaMap.

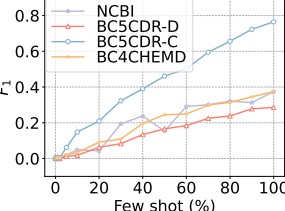

Figure 6: Influence of synonym distance regularizer's weight $\alpha$.

| | SynGen | SynGen w/o NPR+SDR | Label | UMLS |
|---|---|---|---|---|
| manic - depressive illness | ✓ | ✗ | ✓ | ✗ |
| affective disorders | ✓ | ✗ | ✓ | ✗ |
| atrophic benign epidermolysis bullosa | ✓ | ✗ | ✓ | ✗ |
| renal cell carcinoma | ✓ | ✗ | ✓ | ✓ |
| Friedreich ataxia | ✓ | ✗ | ✓ | ✓ |
| progressive optic atrophy | ✗ | ✓ | ✗ | ✗ |
| primary prostate cancer | ✗ | ✓ | ✗ | ✗ |
| nystagmus | ✗ | ✓ | ✗ | ✗ |
| chronic | ✗ | ✓ | ✗ | ✗ |
| human copper overload disease | ✗ | ✓ | ✗ | ✗ |

Table 4: Case Study.

small, the performance increases as the dictionary size increases. Using nearly only 20% dictionary entries can achieve comparable results as using the full set of dictionaries. However, in the MetaMap model, the performance increases linearly as the dictionary size increases. It shows that the word match based model cannot handle the few-shot cases. This observation shows the potency of using our approach to undertake few-shot learning. It should be noted that the performance stops increasing after a certain ratio. This observation can also be explained with Theorem 1. Increasing the dictionary size can only mitigate the second term in Equation (7). After the second term decreases to a certain amount, the first term dominates the error and continues increasing the dictionary size cannot further reduce the upper bound. This observation can further verify the correctness of our theoretical analysis. To further improve the performance, we should consider reducing the first term in Equation (7) and this is how our proposed SDR and NPR work.

**Standard Deviation Analysis.** To further show the effectiveness and consistency of our proposed components, we conduct a standard deviation analysis as shown in Figure 7. We run each model for 10 runs with different random seeds and draw the box plot of the $F_1$ scores. We can see from Figure 7 that our proposed SynGen consistently out-

performs the model variants without the proposed components, which further validates the effectiveness consistency of each component of our SynGen and the effectiveness of the overall SynGen framework.

**Case Study.** We conduct a case study to demonstrate how our proposed NPR and SDR components enhance performance. As shown in Table 4, we select a range of terms from the NCBI corpus. Using SynGen, both with and without the NPR+SDR component, we predict whether each term is a biomedical term. A term predicted as a biomedical term is marked with a check mark (✓); otherwise, it's marked with a cross (✗). Our findings reveal that SynGen accurately identifies certain terms like "maternal UPD 15" as biomedical, even if they are not indexed with the UMLS. However, without the NPR+SDR components, the system cannot recognize such terms, underscoring the significance of these components. Moreover, SynGen is designed to prevent misclassification of common words (e.g., "man", "breast") and peculiar spans like "t (3; 15)" as biomedical entities. Without our proposed components, the system might erroneously categorize them, further emphasizing

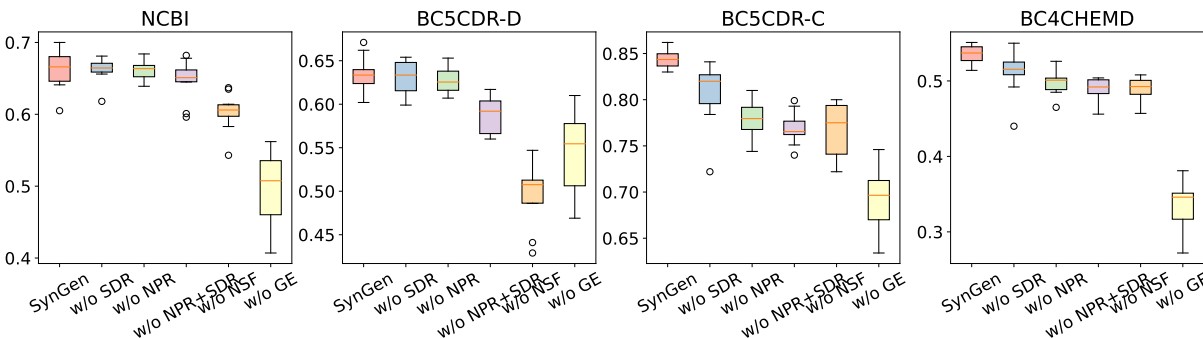

Figure 7: The box plot of each model's $F_1$ score over 10 runs.

the essential role of the NPR+SDR components in SynGen.

## 4 Related Works

Existing NER methods can be divided into three categories, namely, supervised methods, distantly supervised methods, and dictionary-based methods. In supervised methods, Lee et al. (2020) propose to pre-train a biomedical corpus and then fine-tune the NER model. Wang et al. (2019b); Cho and Lee (2019); Weber et al. (2019, 2021) develop several toolkits by jointly training NER model on multiple datasets. However, supervised annotation is quite expensive and human labor-intensive.

In the distantly supervised methods, Lison et al. (2020, 2021); Ghiasvand and Kate (2018); Meng et al. (2021); Liang et al. (2020) propose to first conduct a weak annotation and then train the BioNER model on it. Fries et al. (2017); Basaldella et al. (2020) propose to use utilize a well-designed regex and special case rules to generate weakly supervised data while Wang et al. (2019a); Shang et al. (2020) train an in-domain phase model and make a carefully tailored dictionary. However, these methods still need extra effort to prepare a high-quality training set.

In the dictionary-based methods, Zhang and El-hadad (2013) propose a rule-based system to extract entities. Aronson (2001); Divita et al. (2014); Soldaini and Goharian (2016); Giannakopoulos et al. (2017); He (2017); Basaldella et al. (2020) propose to apply exact string matching methods to extract the named entities. Rudniy et al. (2012); Song et al. (2015) propose to extract the entity names by calculating the string similarity scores while QuickUMLS (Soldaini and Goharian, 2016) propose to use more string similarity scores (Okazaki and Tsujii, 2010) to do a fast approximate dictionary matching. To the best of our knowledge, there is still no existing work that gives a theoretical analysis of the synonyms generalization problem and proposes a corresponding method to solve it.

## 5 Conclusion

This paper proposes a novel synonym generalization framework, i.e. SynGen, to solve the BioNER task with a dictionary. We propose two novel regularizers to further make the terms generalizable to the full domain. We conduct a comprehensive theoretical analysis of the synonym generalization problem in the dictionary-based biomedical NER task to further show the effectiveness of the proposed components. We extensively evaluate our approach on a wide range of benchmarks and the results verify that SynGen outperforms previous dictionary-based models by notable margins.

## Limitations

Although the dictionary-based methods achieve considerable improvements, there is still an overall performance gap compared with the supervised models. Therefore, for domains with well-annotated data, it is still recommended to apply the supervised model. Our proposed SynGen framework is suggested to be applied in domains where there is no well-annotated data.

## Broader Impact Statement

This paper focuses on biomedical named entity recognition. The named entity recognition task is a standard NLP task and we do not make any new dataset while all used datasets are properly cited. This work does not cause any kind of safety or security concerns and it also does not raise any human rights concerns or environmental concerns.

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

# Appendix. Supplementary Material

## A   Proof of Theorem 1

We denote the encoded representation of biomedical named entities $s \in \mathcal{S}$ as $r = E(s) \in \mathcal{P}$ where $|\mathcal{S}| = \mathcal{P}$ and denote the encoded named entities representation in UMLS as $\hat{r} \in \hat{\mathcal{P}}$ where $|\hat{\mathcal{P}}| = |\hat{\mathcal{S}}|$.

**Theorem 1** (Synonym Generalization Error Bound). *If $\hat{\mathcal{S}}$ is an $\epsilon-$net of $\mathcal{S}$. The loss function $f \in [0, b]$ is $\kappa$-Lipschitz continuous. We have with probability at least $1 - \delta$,*

$$E_s < (\kappa\epsilon + b)\sqrt{\frac{\ln|\mathcal{S}| + \ln\frac{2}{\delta}}{2}} + b\sqrt{\frac{\ln\frac{2}{\delta}}{2|\hat{\mathcal{S}}|}}. \tag{7}$$

*Proof.* $\forall r \in \mathcal{P}, \exists \hat{r} \in \hat{\mathcal{P}}$ such that $\|r - \hat{r}\| \leq \epsilon$. We assume that $\forall \hat{r} \in \hat{\mathcal{P}}$ the loss $f(\hat{r})$ is bounded as $f(\hat{r}) \in [0, b]$. Therefore

$$
\begin{aligned}
f(r) &= f(r) - f(\hat{r}) + f(\hat{r}) \\
&\leq \|f(r) - f(\hat{r})\| + f(\hat{r}) \\
&\leq \kappa\|r - \hat{r}\| + f(\hat{r}) \\
&\leq \kappa\epsilon + b
\end{aligned}
$$

On the other hand, as $\hat{\mathcal{P}}$ is evenly sampled from $\mathcal{P}$, namely $\hat{\mathcal{P}} = \{\hat{r}|\hat{r} \sim \mathcal{P}\}$. $\forall f$, we have $\mathbb{E}f(r) = \mathbb{E}f(\hat{r}) = \frac{1}{|\hat{\mathcal{P}}|}\sum_{i=1}^{|\hat{\mathcal{P}}|}\mathbb{E}f(\hat{r}_i) = \mathbb{E}\frac{1}{|\hat{\mathcal{P}}|}\sum_{i=1}^{|\hat{\mathcal{P}}|}f(\hat{r}_i) = \mathbb{E}\hat{R}$

Therefore,

$$
\begin{aligned}
f(r) - \hat{R} &= f(r) - \mathbb{E}f(r) + \mathbb{E}f(r) - \hat{R} \\
&= f(r) - \mathbb{E}f(r) + \mathbb{E}\hat{R} - \hat{R}
\end{aligned} \tag{8}
$$

For the first term $f(r) - \mathbb{E}f(r)$ in Equation (8), by Hoeffding, we have

$$p(f(r) - \mathbb{E}f(r) \geq t) \leq \exp\left(-\frac{2t^2}{(\kappa\epsilon + b)^2}\right)$$

Then, take all $r \in \mathcal{P}$, we have

$$
\begin{aligned}
p(\exists r \in \mathcal{P}|f(r) - \mathbb{E}f(r) \geq t) &= p\left(\bigcup_{r \in \mathcal{P}}\{f(r) - \mathbb{E}f(r) \geq t\}\right) \\
&\leq \sum_{r \in \mathcal{P}} p\left(\{f(r) - \mathbb{E}f(r) \geq t\}\right) \\
&\leq \sum_{r \in \mathcal{P}} \exp\left(-\frac{2t^2}{(\kappa\epsilon + b)^2}\right) \\
&= |\mathcal{P}| \exp\left(-\frac{2t^2}{(\kappa\epsilon + b)^2}\right)
\end{aligned}
$$

Therefore,

$$p(f(r) - \mathbb{E}f(r) < t) = 1 - |\mathcal{P}| \exp\left(-\frac{2t^2}{(\kappa\epsilon + b)^2}\right)$$

Then denote $\delta = |\mathcal{P}| \exp\left(-\frac{2t^2}{(\kappa\epsilon + b)^2}\right)$, we have with probability at least $1 - \delta$,

$$f(r) - \mathbb{E}f(r) \leq (\kappa\epsilon + b)\sqrt{\frac{\ln|\mathcal{P}| - \ln\delta}{2}} \tag{9}$$

Similarly, for the second term $\mathbb{E}\hat{R} - \hat{R}$ in Equation (8), by Hoeffding, we have

$$p(\mathbb{E}\hat{R} - \hat{R} \geq t) \leq \exp\left(-\frac{2|\hat{\mathcal{P}}|^2 t^2}{|\hat{\mathcal{P}}|b^2}\right) = \exp\left(-\frac{2|\hat{\mathcal{P}}|t^2}{r^2}\right)$$

Then denote $\delta = \exp\left(-\frac{2|\hat{\mathcal{P}}|t^2}{b^2}\right)$, we have with probability at least $1 - \delta$,

$$f(\boldsymbol{r}) - \mathbb{E}f(\boldsymbol{r}) \leq b\sqrt{\frac{\ln\frac{1}{\delta}}{2|\hat{\mathcal{P}}|}} \tag{10}$$

By combining Equation (9) and Equation (10), we have

$$p\left(f(\boldsymbol{r}) - \hat{R} < (\kappa\epsilon + b)\sqrt{\frac{\ln|\mathcal{P}| - \ln\delta}{2}} + b\sqrt{\frac{\ln\frac{1}{\delta}}{2|\hat{\mathcal{P}}|}}\right)$$

$$= p\left(f(\boldsymbol{r}) - \mathbb{E}f(\boldsymbol{r}) + \mathbb{E}f(\boldsymbol{r}) - \hat{R} < (\kappa\epsilon + b)\sqrt{\frac{\ln|\mathcal{P}| - \ln\delta}{2}} + b\sqrt{\frac{\ln\frac{1}{\delta}}{2|\hat{\mathcal{P}}|}}\right)$$

$$\geq p\left((f(\boldsymbol{r}) - \mathbb{E}f(\boldsymbol{r}) < (\kappa\epsilon + b)\sqrt{\frac{\ln|\mathcal{P}| - \ln\delta}{2}})\bigcap(\mathbb{E}f(\boldsymbol{r}) - \hat{R} < b\sqrt{\frac{\ln\frac{1}{\delta}}{2|\hat{\mathcal{P}}|}})\right)$$

$$= (1 - \delta)^2$$

$$\geq 1 - 2\delta$$

Therefore, $\forall f, \forall \boldsymbol{r} \in \mathcal{P}$, we have with probability at least $1 - 2\delta$,

$$f(\boldsymbol{r}) - \hat{R} < (\kappa\epsilon + b)\sqrt{\frac{\ln|\mathcal{P}| - \ln\delta}{2}} + b\sqrt{\frac{\ln\frac{1}{\delta}}{2|\hat{\mathcal{P}}|}}$$

Finally, by taking the supremum of the lefthand side terms, we have with probability at least $1 - \delta$,

$$\sup_{\boldsymbol{r}\in\mathcal{P}}(f(\boldsymbol{r}) - \hat{R}) < (\kappa\epsilon + b)\sqrt{\frac{\ln|\mathcal{P}| + \ln\frac{2}{\delta}}{2}} + b\sqrt{\frac{\ln\frac{2}{\delta}}{2|\hat{\mathcal{P}}|}}$$

$\square$

## B  Illustration of SDR and NPR

Figure 8 shows the intuition of how SDR and NPR improve the performance. On the left-hand side, it shows the synonyms and the corresponding loss function value without SDR and NPR. On the right-hand side, it shows the synonym points and the regularized function value. The results indicate that when applied the SDR, the distance between the synonyms for the same concept concentrates more closely with each other. On the other hand, with the NPR, the Lipschitz constant for the function decrease, and the landscape for the function becomes quite flat. As a result, the function value for the synonyms is more close to each other.

## C  Full Results

Table 5 shows the full results of the experimental results.

## D  Hyper-Parameter Tuning

The hyper-parameter with the corresponding search space are listed in Table 6.

## E  Advantages and Disadvantages for NER methods

In Table 7, we compare the advantages and disadvantages of different NER paradigms.

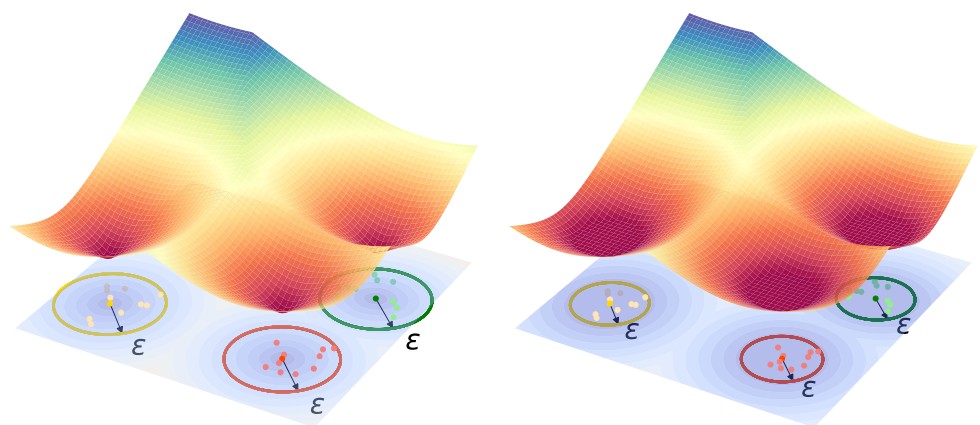

Figure 8: Intuition of SDR and NPR.

| | Model | NCBI | | | BC5CDR-D | | | BC5CDR-C | | | BC4CHEMD | | | Species-800 | | | LINNAEUS | | | AVG | | |
|---|---|---|---|---|---|---|---|---|---|---|---|---|---|---|---|---|---|---|---|---|---|---|
| | | P | R | $F_1$ | P | R | $F_1$ | P | R | $F_1$ | P | R | $F_1$ | P | R | $F_1$ | P | R | $F_1$ | P | R | $F_1$ |
| (Distantly) Supervised | BioBert♮ | 88.2 | 91.2 | 89.7 | 86.5 | 87.8 | 87.2 | 93.7 | 93.3 | 93.5 | 92.8 | 91.9 | 92.4 | 72.8 | 75.4 | 74.1 | 90.8 | 85.8 | 88.2 | 87.5 | 87.6 | 87.5 |
| | SBM♮ | 88.4 | 88.9 | 88.6 | 83.4 | 86.4 | 84.9 | 93.2 | 93.6 | 93.4 | 92.0 | 86.6 | 89.2 | 99.5 | 91.6 | 95.4 | 99.8 | 80.1 | 88.9 | 92.8 | 87.9 | 90.1 |
| | SBMCross◇ | 75.9 | 58.3 | 66.0 | 70.1 | 61.3 | 65.4 | 94.1 | 86.4 | 90.1 | 72.2 | 63.2 | 67.4 | 64.2 | 64.5 | 64.3 | 78.8 | 45.8 | 57.9 | 75.9 | 63.2 | 68.5 |
| | SWELLSHARK♭△ | 64.7 | 69.7 | 67.1 | 80.7 | 77.6 | 79.1 | 88.3 | 88.3 | 88.3 | - | - | - | - | - | - | - | - | - | 77.9 | 78.5 | 78.2 |
| | AutoNER♡♮ | 79.4 | 72.0 | 75.5 | 86.2 | 67.9 | 76.0 | 85.2 | 84.2 | 84.7 | 91.1 | 18.9 | 31.3 | 86.6 | 90.9 | 88.7 | 92.1 | 95.6 | 93.8 | 86.8 | 71.6 | 75.0 |
| | AutoNER w/o DT♡ | 66.8 | 32.4 | 43.6 | 72.0 | 17.3 | 27.9 | 89.7 | 67.3 | 76.9 | 90.7 | 19.7 | 32.4 | 57.6 | 50.7 | 53.9 | 88.4 | 39.0 | 54.1 | 77.5 | 37.7 | 48.1 |
| | AutoNER w/o IDC♮ | 85.1 | 19.1 | 31.2 | 87.1 | 40.4 | 55.2 | 94.2 | 37.3 | 53.4 | 91.2 | 18.8 | 31.2 | 83.6 | 18.5 | 30.3 | 90.4 | 62.8 | 74.1 | 88.6 | 32.8 | 45.9 |
| Dictionary-Based | AutoNER w/o DT+IDC | 57.9 | 9.7 | 16.6 | 63.0 | 13.9 | 22.8 | 92.8 | 39.3 | 55.2 | 60.9 | 24.6 | 35.1 | 59.8 | 25.0 | 35.3 | 80.1 | 33.0 | 46.8 | 69.1 | 24.2 | 35.3 |
| | EmbSim (BioBert) | 56.7 | 24.9 | 34.6 | 61.8 | 14.3 | 23.2 | 71.7 | 61.2 | 66.0 | 47.4 | 24.7 | 32.4 | 49.0 | 34.2 | 40.3 | 80.4 | 42.9 | 55.9 | 61.2 | 33.7 | 42.1 |
| | EmbSim (PubMedBert) | 55.6 | 24.6 | 34.2 | 61.0 | 13.1 | 21.6 | 70.9 | 61.7 | 66.0 | 46.2 | 24.8 | 32.3 | 50.8 | 29.7 | 37.4 | 80.3 | 42.2 | 55.4 | 60.8 | 32.7 | 41.2 |
| | EmbSim (SAPBert) | 58.3 | 27.0 | 37.0 | 61.4 | 14.5 | 23.4 | 71.9 | 61.1 | 66.1 | 46.2 | 24.4 | 31.9 | 50.5 | 28.5 | 36.5 | 80.3 | 42.0 | 55.1 | 61.4 | 32.9 | 41.7 |
| | EmbSim (Word2vec) | 46.2 | 27.6 | 34.5 | 50.2 | 15.3 | 23.5 | 64.9 | 64.8 | 64.9 | 37.9 | 30.4 | 33.7 | 40.6 | 27.2 | 32.6 | 69.4 | 41.7 | 52.1 | 51.5 | 34.5 | 40.2 |
| | MetaMap | 61.8 | 27.8 | 38.4 | 69.3 | 13.3 | 22.3 | 65.9 | 63.5 | 64.7 | 33.1 | 25.2 | 28.6 | 56.9 | 48.7 | 52.5 | 85.5 | 44.3 | 58.3 | 62.1 | 37.1 | 44.1 |
| | MetaMap (Uncased) | 58.4 | 27.5 | 37.4 | 63.5 | 18.4 | 28.6 | **94.8** | 64.1 | 76.5 | **86.2** | 24.0 | 37.5 | 49.1 | 52.3 | 50.6 | 79.1 | 49.6 | 61.0 | 71.9 | 39.3 | 48.6 |
| | SPED | 59.3 | 30.1 | 39.9 | 68.2 | 14.3 | 23.7 | 65.6 | 63.9 | 64.8 | 33.0 | 25.4 | 28.7 | 56.0 | 49.4 | 52.5 | 85.3 | 44.7 | 58.7 | 61.2 | 38.0 | 44.7 |
| | TF-IDF | 26.1 | 29.7 | 27.7 | 32.0 | 22.6 | 26.4 | 74.1 | 65.4 | 69.5 | 19.1 | 39.3 | 25.7 | 42.5 | 21.4 | 28.4 | 77.3 | 40.5 | 53.1 | 45.2 | 36.5 | 38.5 |
| | QuickUMLS (Dice) | 61.2 | 20.9 | 31.2 | 76.1 | 25.2 | 37.8 | 82.0 | 57.8 | 67.8 | 60.1 | 20.6 | 30.7 | 51.9 | 46.7 | 49.1 | 76.6 | 50.2 | 60.6 | 68.0 | 36.9 | 46.2 |
| | QuickUMLS (Cosine) | 52.6 | 22.6 | 31.6 | 62.7 | 32.2 | 42.6 | 74.2 | 58.7 | 65.6 | 47.0 | 22.9 | 30.8 | 44.0 | 46.6 | 45.3 | 67.1 | 50.2 | 57.5 | 57.9 | 38.9 | 45.6 |
| | QuickUMLS (Jaccard) | **80.4** | 17.2 | 28.4 | **93.5** | 14.5 | 25.1 | 93.2 | 56.9 | 70.7 | **82.7** | 16.9 | 28.1 | **61.7** | 46.7 | 53.2 | **88.2** | 44.7 | 59.3 | **83.3** | 32.8 | 44.1 |
| | QuickUMLS (Overlap) | 3.2 | 13.2 | 5.1 | 7.1 | 21.6 | 10.7 | 7.5 | 12.6 | 9.4 | 3.4 | 18.0 | 5.7 | 3.8 | 10.2 | 5.6 | 13.1 | 5.8 | 8.0 | 6.4 | 13.6 | 7.4 |
| | SynGen | 68.8 | **64.1** | **66.2** | 63.8 | **63.4** | **63.5** | 85.0 | **83.9** | **84.4** | 56.4 | **51.1** | **53.6** | 58.8 | **65.7** | **62.0** | 84.9 | **66.2** | **74.4** | 69.6 | **65.7** | **67.4** |
| | SynGen (SAPBert) | 70.0 | 63.2 | 66.3 | 65.4 | 59.3 | 62.1 | 84.4 | 83.2 | 83.8 | 51.0 | 40.7 | 42.4 | 56.7 | 59.5 | 58.1 | 83.0 | 63.0 | 71.6 | 68.4 | 61.5 | 64.0 |
| | SynGen (BioBert) | 59.8 | 65.8 | 62.4 | 58.8 | 56.5 | 57.4 | 83.2 | 79.1 | 81.0 | 45.0 | 30.3 | 31.7 | 55.8 | 49.4 | 52.4 | 82.3 | 46.3 | 59.3 | 64.2 | 54.6 | 57.4 |
| | SynGen (Bert) | 60.1 | 64.2 | 62.0 | 56.5 | 55.6 | 56.1 | 79.4 | 80.0 | 79.6 | 46.4 | 48.2 | 47.1 | 55.1 | 55.4 | 55.2 | 81.3 | 50.2 | 62.1 | 63.1 | 58.9 | 60.4 |
| | SynGen (SciBert) | 57.9 | 66.6 | 61.6 | 56.5 | 55.6 | 55.7 | 83.5 | 78.9 | 81.0 | 55.3 | 45.6 | 49.3 | 54.5 | 51.8 | 53.1 | 80.4 | 52.8 | 63.7 | 64.7 | 58.6 | 60.7 |

Table 5: Main results. We repeat each experiment for 10 runs and report the averaged scores. For BioBert and SWELLSHARK, we report the score from the original paper. We mark the extra effort involved with superscripts, where ♮ is gold annotations; ◇ is in-domain annotations; ♭ is regex design; △ is special case tuning; ♡ is in-domain corpus; ♮ is dictionary tailor. The standard deviation analysis is in Figure 7.

| | Search Space | NCBI | BC5CDR-D | BC5CDR-C | BC4CHEMD | Species-800 | LINNAEUS |
|---|---|---|---|---|---|---|---|
| SDR weight $\alpha$ | {0.05, 0.1, 0.2, 0.5, 1.0, 2.0} | 0.1 | 0.1 | 0.1 | 0.1 | 0.1 | 0.2 |
| NPR weight $\beta$ | {0.05, 0.1, 0.2, 0.5, 1.0, 2.0} | 0.1 | 1.0 | 1.0 | 1.0 | 0.5 | 0.1 |
| NSF minimal distance $t_d$ | {4.0,5.0,6.0,7.0,8.0,9.0,10.0} | 7.0 | 7.0 | 5.0 | 6.0 | 7.0 | 6.0 |
| Inference probability threshold $t_p$ | {0.3,0.4,0.5,0.6,0.7} | 0.5 | 0.6 | 0.5 | 0.5 | 0.6 | 0.5 |
| Inference maximal span length $m_s$ | {6,8,10,12} | 8 | 8 | 10 | 10 | 8 | 8 |

Table 6: Search space for hyper-parameters.

| Paradigm | Advantage | Disadvantage |
|---|---|---|
| Supervised | Best performance. | Demanding large-scale annotated data. |
| Distantly Supervised | No human annotation. | Requiring extra effort, e.g. regex and rule design. |
| Dictionary-based | Only one dictionary is needed. | Poor performance. |

Table 7: Comparison between different NER paradigms.