# OpenReview forum: "Biomedical Named Entity Recognition via Dictionary-based Synonym Generalization"
_EMNLP/2023/Conference — EMNLP 2023 Main_

### Official Review · Reviewer_gSYy · 2023-08-02

**Soundness:** 4

**Excitement:**

4: Strong: This paper deepens the understanding of some phenomenon or lowers the barriers to an existing research direction.

**Paper Topic And Main Contributions:**

This paper investigates biomedical entity recognition. The authors propose a novel dictionary-based approach that does not require manual effort. It does not require a manually annotated training corpus, manually designed rules, or even a task-specific corpus (such as the unlabeled version of a training corpus for a given task). They formulate the task as a synonym generalization problem, i.e. recognizing text spans as synonyms of dictionary concepts. They build an MLP classifier to classify text spans as positive or negative instances, which they train using synonyms from a medical dictionary (the UMLS) as positive examples and text spans extracted from a general biomedical corpus (PubMed) as negative examples. Text spans are encoded using a BERT-based model. The authors propose two regularization steps to improve the generalizability of synonym recognition: one that regularizes the distance between synonyms in the embedding space and one that minimizes the loss gap between synonyms. The authors evaluate their method on 6 BioNER tasks and compare it to several existing supervised, distantly supervised and dictionary-based approaches. They show that their method significantly outperforms all existing dictionary-based approaches.

**Questions For The Authors:**

A) I'm a bit confused about the Greedy Extraction section: the introduction of the section seems to indicate that the issue is to design a method that is able to recognize nested entities. However, the details given afterwards seem to indicate that all nested entities will actually be removed. Could the authors clarify what the method does?

B) It's not clear what dictionary tailoring in the AutoNER approach actually entails. Is that a step that requires extra manual work? If it is fully automatic, then I don't think it is a weakness of the method. As far as I can tell, the most important difference between AutoNER and the proposed SynGen is that AutoNER requires a task specific corpus, while SynGen does not.

**Reasons To Accept:**

- novel and interesting approach that does not require manual annotation, manually designed rules or a task-specific corpus.
- fairly good performance, significantly outperforming other dictionary-based approaches.
- the code is made available

**Reasons To Reject:**

- I'm not sure how accurate the few-shot experiment results (Figure 4) are: at 100%, performance seems lower than what is reported in Table 1, especially for the BC5CDR-D corpus (0.4 vs. 0.635)
- A few details/clarifications are needed: the "Greedy Extraction" part is unclear, and so is the AutoNER approach (see questions below)

**Reproducibility:**

4: Could mostly reproduce the results, but there may be some variation because of sample variance or minor variations in their interpretation of the protocol or method.

**Reviewer Confidence:**

2: Willing to defend my evaluation, but it is fairly likely that I missed some details, didn't understand some central points, or can't be sure about the novelty of the work.

---

### Official Review · Reviewer_JQ58 · 2023-08-05

**Soundness:** 4

**Excitement:**

4: Strong: This paper deepens the understanding of some phenomenon or lowers the barriers to an existing research direction.

**Paper Topic And Main Contributions:**

The paper focuses on the concept synonym identification in the context of biomedical named entity recognition.
The contribution is an approach  which takes advantage of the information issued from dictionary and large  language model to identify concept synonyms.

**Reasons To Accept:**

- the method is well and clearly described.
- experiments are performed on several corpora and the results are compared to state-of-the-art approaches. The system performs better than similar type of methods.
- a theoretical analysis of the synonymy generalization.

**Reasons To Reject:**

- distant supervised approaches achieves better results. But such approaches requires annotated data.

**Reproducibility:**

5: Could easily reproduce the results.

**Reviewer Confidence:**

4: Quite sure. I tried to check the important points carefully. It's unlikely, though conceivable, that I missed something that should affect my ratings.

---

### Official Review · Reviewer_tQnM · 2023-08-07

**Soundness:** 4

**Excitement:**

4: Strong: This paper deepens the understanding of some phenomenon or lowers the barriers to an existing research direction.

**Paper Topic And Main Contributions:**

Paper introduces a novel synonymy generalization framework (SynGen). It recognizes biomedical concepts included in input text using span-based predictions. Paper introduces two regularization methods SDR and NPR to minimize the synonymy generalization error. Extensive evaluations and ablation studies are done to verify the performance of SynGen.

**Reasons To Accept:**

Well written paper with through and carefully designed experiments.

**Reasons To Reject:**

No major reasons to reject.

**Reproducibility:**

4: Could mostly reproduce the results, but there may be some variation because of sample variance or minor variations in their interpretation of the protocol or method.

**Reviewer Confidence:**

3: Pretty sure, but there's a chance I missed something. Although I have a good feel for this area in general, I did not carefully check the paper's details, e.g., the math, experimental design, or novelty.

---

### Official Review · Reviewer_GYVt · 2023-08-11

**Soundness:** 4

**Excitement:**

4: Strong: This paper deepens the understanding of some phenomenon or lowers the barriers to an existing research direction.

**Paper Topic And Main Contributions:**

The authors introduce a dictionary-based biomedical NER framework that’s backed by two important components: a synonym distance regularizer and a noise perturbation regularizer. The authors demonstrated the effectiveness of their methodology against many baselines across multiple datasets.


**Reasons To Accept:**

The authors achieve significant performance improvements against other dictionary-based frameworks across a large number of datasets. Additionally, analysis based on synonym distance, few-shot scenarios based on multiple dictionary sizes, and hyper-parameter optimization (on SDR and NDR) are provided and offer further insight on the effectiveness of SynGen.

The authors provide a thorough and detailed theoretical analysis of their framework.

The paper is clear and concise, making it easy to follow.


**Reasons To Reject:**

Ideally, there would be a case study outlining the effectiveness of SynGen via an example.


**Reproducibility:**

4: Could mostly reproduce the results, but there may be some variation because of sample variance or minor variations in their interpretation of the protocol or method.

**Reviewer Confidence:**

3: Pretty sure, but there's a chance I missed something. Although I have a good feel for this area in general, I did not carefully check the paper's details, e.g., the math, experimental design, or novelty.

---

### Meta-Review · Area_Chair_burv · 2023-09-19

**Recommendation:** 4

**Metareview:**

This paper explores a method for biomedical NER which introduces a synonym generalization framework to improve dictionary-based approaches for this problem. The reviews have minor concerns about the clarity for some details, but the rebuttal has addressed them. The authors are encouraged to update the paper according to the rebuttal and discussion with the reviewers.

---

### Decision · Program_Chairs · 2023-10-07

**Decision:**

Accept-Main

**Comment:**

This paper explores a method for biomedical NER which introduces a synonym generalization framework to improve dictionary-based approaches for this problem. The reviews have minor concerns about the clarity for some details, but the rebuttal has addressed them. The authors are encouraged to update the paper according to the rebuttal and discussion with the reviewers.